# MOLLEO+: Towards Optimized Use of LLMs for Drug Discovery

## Abstract

Large language models (LLMs) have recently emerged as a promising tool for small-molecule generation in drug discovery. One notable recent state-of-the-art work in this field is MOLLEO (Wang et al., 2025), which combines an evolutionary algorithm with an LLM that acts as the operator for making crossovers and mutations on the ligand population. MOLLEO demonstrates strong results on optimizing molecular docking scores, but several aspects of their model are not well suited to real-world drug discovery. We introduce MOLLEO+, an optimized LLM workflow for small-molecule generation. First, we replace docking with the recently released biomolecular foundation model Boltz-2 as an oracle, which improves the predicted binding affinity of generated molecules using gold-standard molecular dynamics by over 100%. Second, we incorporate knowledge of existing ligands, which is present in most practical drug discovery scenarios, using ligands from BindingDB instead of ZINC 250k as the starting population for the genetic algorithm. Third, we propose a fine-tuning strategy to better modify existing ligands towards higher activity. We demonstrate the superiority of MOLLEO+ on the receptor tyrosine kinase c-MET and the BRD4 protein, yielding an improvement over state-of-the-art baselines by up to 20% for Boltz-2 binding affinity.

## 1 Introduction

Large Language Models (LLMs) have recently gained interest for their ability to make significant discoveries and advancements in scientific areas. This is perhaps most notable in the recent AlphaEvolve (Altschul et al., 1990), an evolutionary approach that uses LLMs to progressively improve the quality of a generated algorithm. It successfully developed state-of-the-art algorithms for multiple problems in mathematics and computer science.

However, studies on applying LLMs to the field of small-molecule generation for drug discovery have been limited. Most previous work in machine learning for small-molecule drug design has focused on VAEs (Eckmann et al., 2022; 2025; Noh et al., 2022), diffusion models (Lee et al., 2023; Hoogeboom et al., 2022; Zhou et al., 2024), reinforcement learning (Jeon & Kim, 2020; Fu et al., 2022; Mazuz et al., 2023), and other generative frameworks (Zhu et al., 2023). These methods are often guided by a cheap oracle such as AutoDock (Trott & Olson, 2009), which predicts the binding affinity of generated compounds to a particular protein target; however, it is known to be inaccurate in reflecting actual experimental activity (Handa et al., 2023). Thus, most current frameworks struggle with generating compounds that are likely to show experimental binding.

Recently, LLMs have begun to garner interest as a method to generate small molecule binders, showing promise in generating strong, drug-like ligands. In contrast to more specialized models, LLMs hold the distinct advantage of being implicitly aware of how chemistry is typically done (e.g. common reactions, lead optimization techniques, etc.), giving them great potential in problems related to chemical discovery (White, 2023). This has been demonstrated in the notable previous work MOLLEO (Wang et al., 2025), an evolutionary algorithm that incorporates LLMs as a mutation and crossover operator. They report state-of-the-art results for generating molecules with multiple desired properties, demonstrating the potential of LLMs as a generative framework in the field.

In this work, we further advance LLMs for small molecule drug discovery by introducing a set of novel optimizations to improve their real-world effectiveness. We introduce MOLLEO+, an optimized framework specially designed for optimizing protein-ligand binding affinity. First, we replace

the AutoDock (Trott & Olson, 2009) oracle in MOLLEO with the new biomolecular foundation model Boltz-2 (Passaro et al., 2025). We demonstrate that this relatively cheap oracle significantly improves the quality of generated ligands over AutoDock, as measured by the gold-standard Absolute Binding Free Energy (Feng et al., 2022, ABFE). To our knowledge, this is the first work to demonstrate Boltz-2 as a superior oracle in practice for generative frameworks over the currently standard molecular docking. Second, we change the starting population in MOLLEO to consist of ligands from the large protein-ligand database BindingDB (Liu et al., 2007), focusing the algorithm toward the exploitation of existing strong binders. Third, we construct a semi-synthetic dataset based on BindingDB for improving the LLM's lead optimization capabilities. We use this dataset to fine-tune a small LLM, and significantly improve the quality of its generated ligands.

To summarize, we present MOLLEO+, an optimized framework built upon the state-of-the-art MOLLEO (Wang et al., 2025), greatly improving its performance via the following contributions:

- We replace the docking-based fitness evaluator with Boltz-2 (Passaro et al., 2025) and show that it increases the mean Absolute Binding Free Energy of generated molecules by over 100%.

- We utilize a starting population of ligands based on BindingDB, which increases mean predicted binding affinity of generated compounds by up to 15%.

- We develop a novel post-training framework to fine-tune LLMs for lead optimization tasks using a semi-synthetic dataset. We demonstrate its effectiveness by fine-tuning a small LLM and significantly improving the quality of its generated molecules.

## 2 RELATED WORK

### 2.1 MOLECULAR GENERATIVE MODELS

Most generative models for small-molecule drug design rely on an external oracle that approximates the binding strength of a ligand to a protein target. Graph-GA (Jensen, 2019) is an evolutionary algorithm that keeps track of an active population of molecules, for which their fitness is evaluated by some relevant oracle for the desired optimization property. It executes algorithmic random crossovers and mutations at particular rings and bonds within ligands of the active population, progressively yielding molecules with more desirable properties as the algorithm progresses. Other frameworks have relied on machine learning methods to learn the implicit probability distribution of some input set, ideally generalizing to high performance in the full chemical space. Frameworks like TAGMol (Dorna et al., 2024) and DecomptOpt (Zhou et al., 2024) rely on conditional diffusion models that are further guided toward strong generations by an oracle and some external optimization framework. Pocket2Mol (Peng et al., 2025) employs a graph neural network comprised of several encoder and predictor modules that auto-regressively predicts the location and type of each subsequent ligand atom based on existing ligand atoms and the protein pocket. Notably, Pocket2Mol does not rely on any external oracle in its generation, only the inherent probability distribution learned during training.

Recently, there has been an understanding that common, cheap binding affinity oracles such as AutoDock (Trott & Olson, 2009) are inaccurate in predicting properties that reflect real-life experimental activity (Handa et al., 2023). Physics-based molecular dynamics simulations, e.g. Absolute Binding Free Energy (Feng et al., 2022, ABFE), are currently known to be the most accurate in binding affinity prediction, but they are extremely expensive to run and are thus unrealistic candidates for an oracle that may need to be called on tens of thousands of times within an optimization framework. MF-LAL (Eckmann et al., 2025) is an active learning framework that aims to remedy this problem through a multi-fidelity approach that balances feedback from expensive oracles (e.g. ABFE) and inexpensive oracles (e.g. AutoDock). This results in generations that are more optimal by assessment of the most accurate free energy methods. Ultimately, for the most real-life applicable results from generative models, these high-accuracy molecular dynamics predictors should be of the utmost consideration.

## 2.2 LLM-BASED APPROACHES

Previous work for incorporating LLMs in drug discovery has been relatively limited. There have been efforts in creating specialized models for drug discovery tasks. Models like Y-Mol (Ma et al., 2024) and DrugGen (Sheikholeslami et al., 2025) do this through a combination of pre-training and fine-tuning. However, these models are inherently designed to be one-shot at generation and property prediction, which does not comprise a full and rigorous optimization framework.

The current state-of-the-art in LLM generative frameworks is MOLLEO (Wang et al., 2025), which is a modification of the Graph-GA algorithm (Jensen, 2019). It utilizes LLMs to make structural modifications (crossovers and mutations) to the ligand population. This incorporation of LLMs into the algorithm resulted in strong results for protein-ligand optimization on 3 protein targets, demonstrating the potential of these inherently chemistry-aware large models to be a competitive generative framework in drug discovery. Due to the use of a natural language model as the operator, MOLLEO employs a string representation of small molecules, using both the Simplified Molecular Input Line Entry System (Weininger, 1988, SMILES) and Self-Referencing Embedded Strings (Krenn et al., 2020, SELFIES) for various LLMs. We focus on SMILES in this work.

While previous models mentioned above have succeeded in fine-tuning more chemistry-aware LLMs, the intended one-shot nature of their generations make them unsuitable for use in a multi-step optimization process such as MOLLEO. In contrast, fine-tuning framework we introduce in this work is optimized for conditioned generation, in which the LLM needs to generate a molecule based on provided information of previous members in the ligand population. In this way, our fine-tuning approach is more narrow and focused than previous work, meant for particular use within long optimization frameworks.

## 3 METHODOLOGY

**Problem Statement**    Formally, we can represent our molecular optimization problem as

$$m^* = \underset{m \in M}{\operatorname{argmin}} \, \Phi_p(m)$$

where $m$ is any valid molecule (ligand) and $M$ is the entire valid chemical space. $\Phi_p : M \to \mathbb{R}$ is an evaluation function that predicts the scalar binding free energy ($\Delta G$, in kcal/mol) of $m$ to the protein binding target $p$. Lower binding free energy indicates stronger binding. We aim to find the optimal molecule $m^*$, which minimizes the evaluation function (also called the oracle).

MOLLEO is a genetic algorithm that uses an LLM to generate offspring based on modifications of ligands in the population. Each offspring is subsequently evaluated by the function $\Phi_p$. The molecules with the highest "fitness" according to $\Phi_p$ are chosen for the next population. The LLM performs both crossover and mutation operations, only falling back on algorithmic modifications if the LLM fails to generate a valid molecule. We now discuss the 3 main contributions that comprise MOLLEO+, which are visually represented in Figure 1.

### 3.1 BOLTZ-2 AS A ORACLE

Boltz-2 (Passaro et al., 2025) is a new biomolecular foundation model that utilizes a transformer-based, SE(3) equivariant architecture to carry out 3D structure prediction, and subsequent binding affinity estimation on the predicted structure. The authors show that Boltz-2 approaches the accuracy of much more expensive gold-standard free energy methods like Absolute Binding Free Energy (ABFE) on their evaluation set, at around 1/1000 the cost. Notably, it is the first pure deep learning model to approach this kind of accuracy.

In this work, we replace the docking-based reward function used in MOLLEO with the more accurate affinity predictions from Boltz-2, which adds minimal computational cost. Formally, we change the original MOLLEO evaluation function $\Phi_p$ from AutoDock to Boltz-2 for any protein target $p$ with a known amino acid sequence. In other words, we directly utilize Boltz-2 affinity prediction to assign a fitness score to every generated ligand offspring. To our knowledge, this is the first work to demonstrate the advantages of using Boltz-2 as an oracle within a generative framework.

Figure 1: **The 3 primary optimizations that comprise MOLLEO+.** (1) We employ Boltz-2 as an oracle for improved ABFE results. (2) We utilize BindingDB to form a much stronger basis for optimization. (3) We propose a novel fine-tuning framework to steer LLMs toward stronger molecule generations.

## 3.2 OPTIMIZING STARTING POPULATION OF MOLLEO

The strength and diversity of the starting population for a genetic algorithm is crucial for the quality of generated compounds, because all ligand offspring are derived in some way from the structures present in the starting population. The original MOLLEO algorithm uses a random sample of ZINC 250k (Sterling & Irwin, 2015) compounds as the initial population. Although this data set does provide a diverse pool of structures to build on, the molecules present are inherently not designed for any particular target.

Our optimization to MOLLEO involves employing the large protein-ligand database BindingDB (Liu et al., 2007) instead of ZINC 250k to give the MOLLEO algorithm a significantly stronger starting point. With BindingDB, we are able to selectively pick strong known binders to the particular target that we are interested in, comprising an initial population that immediately promises much greater experimental activity. This focuses the algorithm more on the exploitation of existing strong binders (which are often known during drug discovery projects), rather than exploration based on non target-specific molecular structures.

To form this starting pool, we first separate the set of BindingDB ligands corresponding to the protein target we want to target into clusters using the Butina algorithm, which creates clusters based on the pairwise Tanimoto similarity of all ligands to each other. We use a distance threshold of 0.4. This ensures that ligands are structurally diverse across different clusters, because very similar ligands are all grouped within the same clusters. This is desirable because we want the algorithm to have access to a diverse set of structures and molecules, giving it the potential to create entirely novel molecules through combinations and crossovers. From there, we sampled the ligand with the best binding affinity from each cluster, forming a set of strong-binding, structurally diverse ligands. After sorting this list of ligands, we provide the top $n$ ligands with the best binding affinity as the starting population for MOLLEO. Analysis of the diversity of the BindingDB starting population in comparison to the ZINC 250k starting population can be found in Appendix B

## 3.3 FINE-TUNING WITH BINDINGDB

We propose a fine-tuning strategy to imbue domain knowledge and specifically enable an LLM to perform better at each optimization step of the MOLLEO process, i.e. every time the LLM makes a crossover/mutation based on previous ligands. We can formulate the process of each individual optimization step $i$ as $m_i = LLM(h_{<i})$ where $h_{<i}$ is the prior information (SMILES and binding affinity) given about previous ligands in the population, and $m_i$ is the newly generated molecule. We want to tune an LLM that can most effectively process $h_{<i}$ to yield $m_i$ that minimizes the binding affinity evaluation function $\Phi_p$. To achieve this, we propose a supervised fine-tuning (SFT)

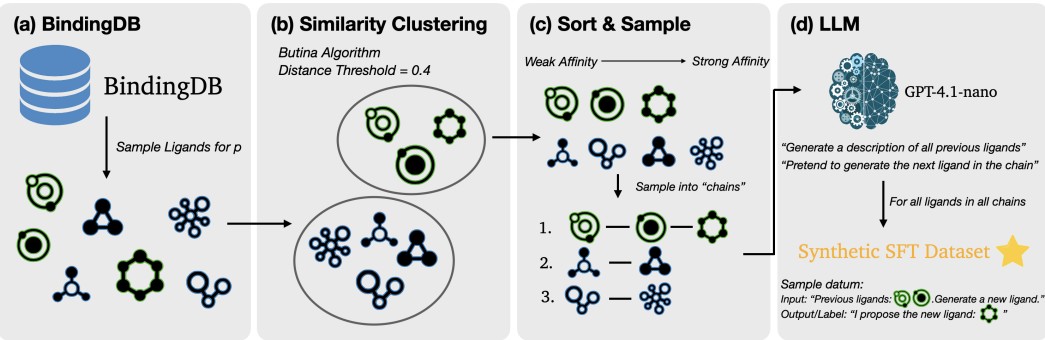

Figure 2: **Pipeline for preparing SFT dataset**. (a) We sample ligands for the desired protein target $p$ from BindingDB. (b) We cluster the ligands by the Butina algorithm. (c) We sort the ligands by binding affinity within each cluster, then sample them into several chains. (d) We utilize a LLM to generate our semi-synthetic dataset from the ligand chains.

framework for any particular protein target $p$, involving the creation of a semi-synthetic dataset followed by subsequent usage of the dataset in SFT.

To form a semi-synthetic dataset for supervised fine-tuning using BindingDB, we begin in a similar way to the clustering method described above for the formation of the BindingDB starting population. We form $n$ distinct clusters from the ligand pool for our desired protein target, using Butina clustering with distance threshold 0.4. Then within each cluster, we first sort the ligands by affinity, then form a series of "ligand chains". This is done by first picking a weak affinity ligand, then repeatedly selecting a ligand with binding affinity strictly stronger than the current. The result is that for each cluster, we end up with several chains of ligands that are ordered with increasing binding affinity. All ligands within a chain are guaranteed to be relatively similar in structure due to the clustering. The goal of this approach is to form a dataset where an LLM learns to make decisions that change a weak-binding ligand into a guaranteed strong-binding one as it moves down the chain during training. The changes are usually minimal due to the structural similarity, so each chain represents a somewhat realistic series of modifications that a medicinal chemist might make. An example of one of these ligand chains generated for the c-MET target can be found in Appendix C.1.

Employing the ligand chains, we generate a semi-synthetic text dataset using an LLM. We employ GPT 4.1 nano (OpenAI et al., 2024) for cost efficiency. For each ligand chain, we have the LLM generate an artificial input and output response that mimics how we want our tuned LLM to generate a molecule based on previous molecules. Formally, for any ligand in position $i$ of a chain, we create our prior information $h_{0...i}$ by utilizing the LLM to summarize the information from all previous ligands in positions 0 to $i$. Since we want our tuned model to generate the ligand at position $i$, we utilize the LLM to generate a sample output that includes ligand $i$ as the final generation. Then a full input/output datum for the dataset is comprised of the prior $h_{0...i}$ as the input and a sample output that includes the desired molecule $m_i$. Fine-tuning an LLM on this dataset guides the LLM to generate the strong molecule $m_i$ given prior information $h_{0...i}$, which is what we want the model to do within a long optimization process such as MOLLEO. The exact details and prompts for how we utilize the ligand chains to form an SFT dataset can be found in Appendix C.2. This entire process is visually demonstrated in Figure 2.

This semi-synthetic dataset is used in a classic supervised fine-tuning run. We employ a train-validation split on the dataset, and progress until we observe the validation loss reach a plateau. We apply this training framework to the relatively small Llama-3.1-8B-Instruct model (Grattafiori et al., 2024). Details about the SFT training process are provided in Appendix C.3

# 4 RESULTS

## 4.1 IMPROVING ABFE WITH BOLTZ-2 ORACLE

We first evaluate the impact of using Boltz-2 as an oracle instead of AutoDock. Table 1 compares the mean Absolute Binding Free Energy (ABFE) scores (Feng et al., 2022) of ligands generated for the c-MET protein target using Boltz-2 (Passaro et al., 2025) and AutoDock docking (Trott & Olson, 2009) as the fitness evaluator for MOLLEO. We also report the 1st, 2nd, and 3rd strongest molecules generated by these methods. Our setup for ABFE calculations is provided in Appendix A. We also benchmark against molecules generated by MF-LAL (Eckmann et al., 2025), a VAE-based generative method that focuses on achieving strong ABFE results using a multi-fidelity approach.

Table 1: c-MET ABFE results (kcal/mol) for Autodock, Boltz-2, and MF-LAL

| Method | Count | Mean $\pm$ SD | 1st | 2nd | 3rd |
|---|---|---|---|---|---|
| MF-LAL | 10 | -4.3 $\pm$ 3.7 | -8.7 | -8.5 | -8.3 |
| MOLLEO (AutoDock) | 20 | -3.8 $\pm$ 4.2 | -12.8 | -8.8 | -8.7 |
| MOLLEO (Boltz-2) | 20 | **-8.7** $\pm$ 4.6 | **-15.64** | **-14.04** | **-13.98** |

For these calculations, we take the top 20 best molecules generated from each run according to the respective oracle. We can see that MOLLEO does not beat the MF-LAL baseline in terms of mean affinity, but simply incorporating Boltz-2 as the evaluation function improves the results drastically. MOLLEO with Boltz-2 results in compounds with much better ABFE scores than with AutoDock, having a difference in mean ABFE score of -4.8 kcal/mol, a percentage increase of over 100%. MOLLEO with Boltz-2 yields $p = 0.0007$ from the one-sided independent Student's t-test against MOLLEO with AutoDock, and $p = 0.007$ against MF-LAL. Note that these runs are done on the base MOLLEO setup, without the BindingDB starting population described previously.

Given the demonstrated advantages of using Boltz-2 within a generative framework, we are motivated to provide further general analysis of the correlation between Boltz-2, AutoDock, and ABFE. In Figure 3, we take 32 compounds for c-MET, 16 of which are known binders, and 16 of which are presumed inactive binders. We calculate the ABFE, Boltz-2, and AutoDock binding affinities for all 32 compounds. We exclude results for any failed AutoDock or Boltz-2 runs.

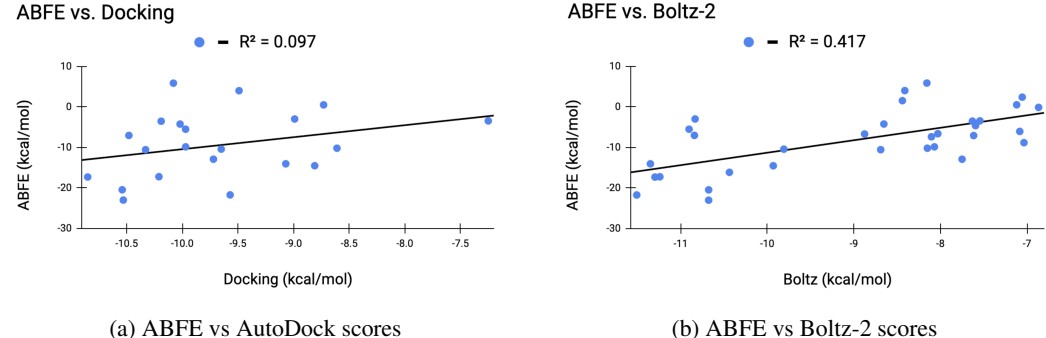

(a) ABFE vs AutoDock scores                    (b) ABFE vs Boltz-2 scores

Figure 3: Comparison of correlation between AutoDock & ABFE and Boltz-2 & ABFE for 32 known compounds for the c-MET protein target. We observe a significantly higher correlation between Boltz-2 and ABFE as compared to AutoDock.

We see that ABFE and AutoDock docking show $r^2 = 0.09$ among the 32 compounds, while ABFE and Boltz-2 show $r^2 = 0.42$. As an oracle nearly 1000x less computationally expensive than ABFE, Boltz-2 shows exceptional correlation with ABFE, especially in comparison to docking. Furthermore, we calculate the ROC-AUC score for Boltz-2 and docking, to see how well they can separate binders from non-binders. Boltz-2 scores 0.95 for this metric, while AutoDock scores 0.84. Due

to computational and time constraints regarding expensive ABFE calculations, we are only able to provide results for the c-MET target at this time.

Thus, we demonstrate that not only does Boltz-2 have stronger correlation with the most accurate gold-standard computational methods, but that it also has practical application within a generative framework, acting as a more accurate evaluator that guides generated compounds towards higher ABFE scores. We generally observe Boltz-2 to be approximately a factor of 10 more expensive to run than AutoDock; however, this difference is entirely negligible in comparison to the cost of molecular dynamics methods such as ABFE.

## 4.2 IMPROVED BINDING AFFINITY WITH MOLLEO+

Next, we demonstrate the performance of MOLLEO+ on two protein targets, c-MET and BRD4. Structurally speaking, these targets are quite dissimilar, making our results more robust when considering both targets. Every MOLLEO run terminates at 1000 oracle calls, with an initial population (and population size) of 120 and offspring size of 70. We report all results with Boltz-2 calculated binding affinities instead of ABFE due to computational constraints, but rely on the strong results shown in the previous section to support the validity of *relative* differences between methods.

Table 2 shows the comparison of binding affinity measured by Boltz-2 across different methods. We compare with the base MOLLEO algorithm, as well as with two additional baselines: MF-LAL and Pocket2Mol (Peng et al., 2025). Both models have previously been observed to yield molecules with high ABFE scores (Eckmann et al., 2025).

We report 4 metrics:

1. Mean and standard deviation in binding affinity (kcal/mol), as predicted by Boltz-2.

2. Number of ligands that exceed a strong-binding threshold of -11 kcal/mol.

3. Quantitative estimate of drug-likeness (Bickerton et al., 2012, QED), a scale from 0-1 for which higher QED indicates higher drug-likeness.

4. Synthetic accessibility (Ertl & Schuffenhauer, 2009, SA), a scale from 1-10 for which lower SA indicates greater ease of molecular synthesis.

Table 2: Boltz-2 affinity (kcal/mol), QED, and SA for baselines and MOLLEO+

| Method | c-MET | | | | BRD4 | | | |
|---|---|---|---|---|---|---|---|---|
| | Mean ± SD | # Strong | QED | SA | Mean ± SD | # Strong | QED | SA |
| MF-LAL | -7.4 ± 1.2 | 0 | **0.56** | 3.7 | -8.8 ± 1.2 | 0 | **0.57** | 3.7 |
| Pocket2Mol | -11.2 ± 0.3 | 7 | 0.38 | 4.7 | -10.2 ± 0.5 | 1 | 0.36 | 4.5 |
| MOLLEO | -9.0 ± 0.4 | 0 | 0.17 | 4.0 | -8.5 ± 0.6 | 0 | 0.19 | 4.5 |
| MOLLEO (BindingDB) | -10.2 ± 0.5 | 4 | 0.20 | 3.8 | -9.2 ± 0.3 | 0 | 0.12 | 4.2 |
| MOLLEO (Boltz-2) | -11.2 ± 0.1 | 9 | 0.18 | 4.8 | -10.7 ± 0.1 | 2 | 0.12 | 4.9 |
| MOLLEO+ (ours) | **-11.9** ± 0.1 | **10** | 0.20 | 4.5 | **-11.9** ± 0.2 | **10** | 0.36 | 3.5 |
| MOLLEO+ (Llama) | -10.9 ± 0.1 | 4 | 0.39 | 4.0 | -10.7 ± 0.1 | 2 | 0.43 | 3.8 |
| MOLLEO+ (Llama FT) | -11.6 ± 0.1 | **10** | 0.35 | **3.7** | -11.4 ± 0.1 | **10** | 0.45 | **3.3** |

MOLLEO is the original, unmodified algorithm. MOLLEO (BindingDB) is the result of utilizing the BindingDB starting population while keeping AutoDock as the oracle. MOLLEO (Boltz-2) is the result of changing the oracle to Boltz-2 instead of AutoDock while keeping the original ZINC 250K starting population. MOLLEO+ is our full method, the result of combining both the Boltz-2 and the BindingDB starting population optimizations. All MOLLEO runs are done with GPT-4.1-mini for cost efficiency, as opposed to GPT-4 used in the original MOLLEO paper. At the bottom, we give the results of using the Llama-3.1-8B-Instruct model within MOLLEO, as well as with our fine-tuned version of this model. For each result, we run the same method 3 times with different randomization seeds for greater rigor.

All metrics are reported for a sample of generated ligands. We first Butina cluster the full pool of generated molecules (with similarity threshold = 0.6), then take the best 10 scores that belong to

distinct clusters. This way, we more effectively assess the quality of structurally unique generations. Additionally, to focus solely on the LLM performance, we remove compounds from MOLLEO runs generated by the default crossover/mutation operators, which the algorithm falls back on if the LLM generates an invalid molecule.

In terms of mean affinity, we see that the original MOLLEO algorithm does not significantly outperform the non-MOLLEO baselines on either target. However, we observe a substantial increase in affinity both when we utilize the BindingDB starting population ($p = 0.03$ against the original MOLLEO for c-MET, $p = 0.12$ for BRD4) and when we use Boltz-2 as the oracle ($p < 0.001$ against the original MOLLEO for both targets) independently of each other. Furthermore, MOLLEO+ yields another significant increase in mean affinity over either previous method by itself, with the combination of both BindingDB and Boltz-2 having significant effects on the affinity results. ($p < 0.001$ against MOLLEO with Boltz-2 for both targets). We also observe a significant increase in affinity between our fine-tuned small Llama model and the untuned version ($p < 0.001$ for both targets). We further measure the number of diverse, top compounds that exceed an activity threshold of -11 kcal/mol; we see that incorporating BindingDB in the starting population increases this metric significantly, as does the fine-tuning process for the Llama model.

We observe that, in general, QED metrics are weaker in MOLLEO+ runs than within the baselines (although, notably, we observe strong results with our fine-tuned Llama model). We note that the original MOLLEO had a similar issue in this area, showing similar if not even weaker QED and SA results compared to MOLLEO+. As an evolutionary algorithm, MOLLEO can support multi-objective optimization through Pareto front optimization. We did not explore multi-objective optimization in this work as our focus was on protein-ligand binding affinity and translation to real-world experimental activity; however, as MOLLEO+ does not suffer any observed decrease in other desirable metrics such as QED and SA as compared to MOLLEO, we expect to still see strong results in other properties through application of the multi-objective framework to our method. In other words, we observe a clear increase in binding affinity in MOLLEO+ with no decrease in desirable ligand-specific properties, implying an unconditional improvement over the original algorithm and a preserved potential for strong performance in multi-objective optimization. We aim to explore this concretely in future work.

### 4.3 ANALYSIS OF SIMILARITY AND NOVELTY

A significant part of the molecular generation process is producing molecules that are both strong binders and relatively novel in structure. We analyze the effectiveness of MOLLEO+ in this area by applying a filter to the results, only considering generated ligands with Tanimoto similarity < 0.5 to *any* ligand in the starting population. The results of applying this filter are shown in Table 3. We provide the average maximum similarity to any ligand in the starting population for generated ligands, then measure the mean of the top 10 diverse ligands with this filter applied.

Table 3: Comparison of average maximum similarity and filtered mean affinity (kcal/mol)

| Method | c-MET | | BRD4 | |
|---|---|---|---|---|
| | Avg. Max Sim. | Filtered Mean ± SD | Avg. Max Sim. | Filtered Mean ± SD |
| MOLLEO | **0.33** | -9.0 ± 0.4 | **0.32** | -8.5 ± 0.6 |
| MOLLEO (Boltz-2) | 0.34 | -11.2 ± 0.1 | **0.32** | -10.7 ± 0.1 |
| MOLLEO+ (ours) | 0.39 | **-11.7** ± 0.1 | 0.40 | **-11.5** ± 0.3 |
| MOLLEO+ (Llama) | 0.36 | -10.2 ± 0.6 | 0.44 | -8.7 ± 0.7 |
| MOLLEO+ (Llama FT) | 0.44 | -11.2 ± 0.3 | 0.37 | -11.0 ± 0.2 |

We observe that MOLLEO+ generally suffers from slightly lower novelty within its generated pool of molecules. This is a consequence of utilizing strong structures in BindingDB as the starting population. Because the provided initial structures are already very strong, the strongest generated ligands are likely to be close derivatives of those initial structures. Then as the evolutionary algorithm progresses, it selects for the strongest ligands, resulting in higher similarity throughout the population. In contrast, the starting structures in ZINC 250K are not guaranteed to be strong for any particular protein target, so for novel generated ligands to be strong, they necessarily have to be

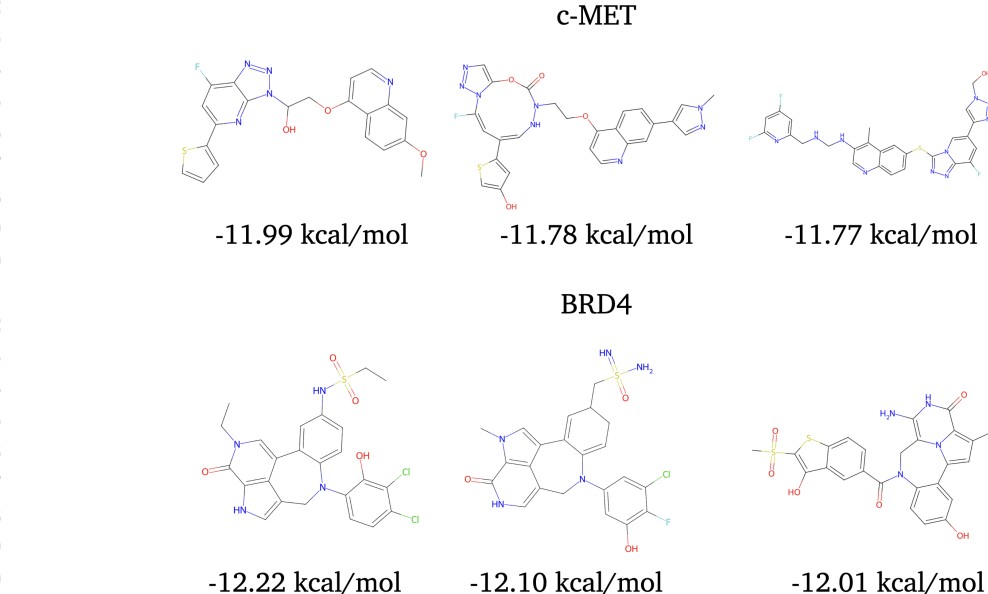

Figure 4: Visualizations of top 3 ligands for c-MET and BRD4 generated by MOLLEO+ with $< 0.5$ Tanimoto similarity to any ligand in the starting population and $< 0.6$ similarity to each other.

dissimilar to much of the starting population. Because of this, we observe higher similarity on average in MOLLEO+. However, the filtered mean shows us that even with a less novel ligand pool *on average*, the novel generations of MOLLEO+ with BindingDB are still significantly stronger than the novel generations of MOLLEO with ZINC 250K ($p = 0.01$ against the Boltz-2 run for c-MET, $p = 0.03$ for BRD4). Visualizations of the top ligands for c-MET and BRD4 when applying the similarity filter are shown in Figure 4.

For extended rigor, we further test filters using more constraining Tanimoto similarity thresholds of 0.45 and 0.4. Additionally, we test a filter based on scaffold-level similarity (at similarity threshold 0.5) by evaluating the similarity in the Murcko Scaffolds generated by ligands instead of full-molecule Tanimoto similarity. Mean affinity for all three of these filters are shown in Table 4 on MOLLEO (Boltz-2) and MOLLEO+ results, for which the comparison between the two is most pertinent to further rigor, since we are most interested in how introducing BindingDB in particular impacts generation diversity. We report the filtered binding affinity means as well as the independent Student's t-test p-values for the alternative hypothesis: MOLLEO+ mean affinity $<$ MOLLEO (Boltz-2) mean affinity.

Table 4: Affinity results after applying 3 separate filters: similarity threshold = 0.45, similarity threshold = 0.4, and Murcko scaffold similarity (with threshold = 0.5)

| Method | c-MET (Mean Affinity) | | | BRD4 (Mean Affinity) | | |
|---|---|---|---|---|---|---|
| | Sim = 0.45 | Sim = 0.4 | Scaffold-level Sim. | Sim = 0.45 | Sim = 0.4 | Scaffold-level Sim. |
| MOLLEO (Boltz-2) | $-11.2 \pm 0.1$ | $-11.2 \pm 0.1$ | $-11.2 \pm 0.1$ | $-10.7 \pm 0.1$ | $-10.7 \pm 0.1$ | $-10.7 \pm 0.1$ |
| MOLLEO+ (ours) | $\mathbf{-11.6 \pm 0.1}$ | $\mathbf{-11.3 \pm 0.1}$ | $\mathbf{-11.6 \pm 0.1}$ | $\mathbf{-11.3 \pm 0.2}$ | $\mathbf{-11.0 \pm 0.1}$ | $\mathbf{-11.3 \pm 0.1}$ |
| T-test P-value | $< 0.01$ | 0.08 | 0.02 | 0.04 | 0.06 | $< 0.01$ |

We observe that, under these additional filters, the incorporation of BindingDB still yields stronger molecules constrained to the necessity for structural diversity. This is true even when evaluating more fine-grained scaffold-level similarity. However, at similarity thresholds lower than 0.4, the gain in affinity from introducing BindingDB becomes minimal. Due to this, we acknowledge that the overall diversity of strong results from MOLLEO+ is generally weaker than when that of using a ZINC 250K starting population. However, given that our primary motivation for this work is to create a framework that takes advantage of known information to create molecules suitable for real-

world drug discovery, we do not consider this an imminently crucial problem. Although the highest possible generation diversity is certainly desirable, we believe that the demonstrated gains in affinity through several levels of restrictive similarity filters show that MOLLEO+ performs well enough in novelty to support its improvement over MOLLEO as a tool to assist drug-discovery pipelines, particularly for lead-optimization tasks where we have prior information and novelty is not of utmost importance. In other words, we believe that we currently demonstrate strong-enough results in this section to justify the larger intended use-case for our framework, which involves generation of strong, relatively novel real-world candidate molecules that are optimized for high scores on the most accurate free-energy benchmarks. We plan to utilize multi-objective optimization to further strengthen novelty results in future work.

## 5 DISCUSSION & CONCLUSION

We present MOLLEO+, an optimization of the MOLLEO framework that demonstrably improves the design of LLM-based protein-ligand drugs. We show that Boltz-2 is a better fitness function than docking, producing compounds that are more likely to show real-world binding. This result is notable given previous concerns that Boltz-2 performs poorly out-of-distribution, and suggests that Boltz-2, instead of docking, should be used as an oracle for other molecular generative models. We also modify the starting population of MOLLEO, resulting in significantly stronger generated structures and molecules throughout the algorithm. Finally, we present a fine-tuning framework that employs BindingDB to create a novel semi-synthetic dataset, which improves the molecular generation abilities of a small Llama 3 model.

We demonstrate significant advantages of the MOLLEO+ framework in generating molecules with high Boltz-2 scores. Due to the demonstrated increase in ABFE and the additional correlation analysis between ABFE and Boltz-2, we are confident that this increase in Boltz-2 affinity over MOLLEO is somewhat correlative to an increase in ABFE. Through this, we are comfortable with claiming that MOLLEO+ is a significantly more effective framework for producing molecules that may show real-world experimental activity.

Our fine-tuning method also shows great promise in improving the generation quality of small molecules through post-training alone. We observe not only a significant increase in binding affinity, but also great success in QED and SA. While we acknowledge that we do not yet exceed the state-of-the-art (GPT-4) in terms of binding affinity with our very small fine-tuned model, we demonstrate extremely strong relative improvements, and hypothesize that the same post-training method can be applied to larger models and yield a similar relative increase in performance. We leave this for exploration in future work. We did have some specific concerns about whether this fine-tuning process actually improved the *quality* of generated molecules, or simply molded the model to obey the MOLLEO answer format. We observed that the untuned model fails to follow the answer format for a significant portion of its generations, leading to failed parsing and thus an unconsidered, potentially-valid generated molecule. We ablate this potential flaw in Appendix C.4.

**Limitations** While our BindingDB approach to MOLLEO demonstrably improves the performance on the c-MET and BRD4 targets, we recognize that these are both very well-studied targets. For less studied protein targets, this method may be entirely inapplicable if there are not enough *diverse* known binders to comprise a starting population. This also somewhat applies to our fine-tuning framework; however, we demonstrate in Appendix D that we can utilize other ligands from BindingDB and still achieve comparable performance.

**Ethics Statement** We recognize that improved molecular optimization frameworks may be utilized to generate chemically dangerous compounds. However, since our work does not consider complicated properties and requirements for generation and synthesis of harmful compounds, our contribution is not imminently problematic in this direction.

**Reproducibility Statement** We provide all details of our work and implementation in the Methodology section, as well as in sections of the Appendix. We disclose our parameters for ABFE, our parameters for fine-tuning, and all details needed to reproduce our experiments. Additionally, we attach all of our relevant code as supplementary material in this submission, which is documented with instructions on how to run everything and reproduce our results.

**LLM Usage**   While our methodology focuses on the use of LLMs as a scientific tool, we did not employ any LLMs in either the ideation or composition of this work. All text in this work were written solely by the authors, as are all figures, tables, and data. We do not employ any LLM assistance in the writing process, and all written text is our own.

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

## A ABFE Setup

For our ABFE calculations, we utilize the following Binding Affinity Tool BAT.py Heinzelmann & Gilson (2021) repository: https://github.com/GHeinzelmann/BAT.py. We simulate using OpenMM and the standard SDR method. For calculations of molecules generated by docking as the oracle, we use the ligand pose generated by AutoDock as the starting pose for the calculation. For calculations of molecules generated by Boltz-2 as the oracle, we use the Boltz-2 predicted ligand pose as the starting pose. We separate the source of the poses to avoid potential bias toward one particular oracle in the ABFE calculation. Because Boltz-2 does not take a protein crystal structure as input and makes a prediction based on the given amino acid sequence, we first align the entire predicted Boltz-2 conformation to the protein crystal structure with ChimeraX Pettersen et al. (2020), then extract only the ligand pose for ABFE. We observe this alignment to yield an RMSE of under 0.7 angstroms; thus we are comfortable using the aligned ligand pose with the crystal structure in ABFE calculations. We do not observe frequent steric clashes resulting from this process.

Our simulation steps parameters for the BAT.py framework are as follows:
eq_steps1 = 500000 (Number of steps for equilibration gradual release)
eq_steps2 = 15000000 (Number of steps for equilibration after release)

m_steps1 = 500000 (Number of steps per window for component m (equilibrium))
m_steps2 = 1000000 (Number of steps per window for component m (production))

n_steps1 = 500000 (Number of steps per window for component n (equilibrium))
n_steps2 = 1000000 (Number of steps per window for component n (production))

e_steps1 = 250000 (Number of steps per window for component e (equilibrium))
e_steps2 = 500000 (Number of steps per window for component e (production))

v_steps1 = 500000 (Number of steps per window for component v (equilibrium))
v_steps2 = 1000000 (Number of steps per window for component v (production))

On 4 NVIDIA H200 GPUs, one ABFE calculation typically takes us around 16 hours to complete.

## B  DIVERSITY IN BINDINGDB STARTING POPULATION

We very briefly analyze the diversity within the BindingDB starting population for c-MET and BRD4, in comparison to the diversity of the ZINC 250k starting population. In Table 5, we calculate the mean pairwise Tanimoto similarity for ZINC 250K, BindingDB for c-MET, and BindingDB for BRD4. Because the ZINC 250K starting population is just a random sample of 120 ligand from the entire set, we randomly sample from ZINC 250K 100 times to reduce variance.

Table 5: Starting population diversity for ZINC 250K vs BindingDB

| Starting Population | Mean Pairwise Tanimoto Similarity |
| --- | --- |
| ZINC 250K | 0.15 |
| BindingDB (c-MET) | 0.20 |
| BindingDB (BRD4) | 0.19 |

We observe a very minimal difference in mean pairwise similarity between ZINC 250K and the BindingDB samples. This indicates that our method involving Butina clustering is sufficient in creating a pool of molecules that are diverse enough for use as a starting population in MOLLEO.

## C  ADDITIONAL LLM FINE-TUNING INFORMATION

### C.1  EXAMPLE LIGAND CHAIN ANALYSIS

In this section, we briefly analyze a generated ligand chain for the c-MET target to provide furter insight and transparency on the construction of the SFT dataset. The full chain of SMILES and corresponding experimental affinity is as follows:

1. COc1cc2c(Oc3ccc(NS(=O)(=O)c4cccc(-c5cnn(C)c5)c4)cc3F)ccnc2cc1OCCCN1CCOCC1 -7.859743051032407

2. COc1cc2c(Oc3ccc(NS(=O)(=O)c4cccc(-c5cnn(C)c5)c4)cc3F)ccnc2cc1OCCCN1CCN(C)CC1 -8.519634173245192

3. COc1cc2c(Oc3ccc(NS(=O)(=O)c4c(F)cccc4F)cc3F)ccnc2cc1OCCCN1CCN(C)CC1 -9.40241538421135

4. COc1cc2c(Oc3ccc(NC(=O)C4(c5nnc(-c6ccc(F)cc6)o5)CC4)cc3F)ccnc2cc1OCCCN1CCCCC1 -9.746740922094766

5. COc1ccc(/C=N/NC(=O)Nc2ccc(Oc3ccnc4cc(OCCCN5CCOCC5)c(OC)cc34)c(F)c2)cc1 -9.88726911173949

6. CCC(=NNC(=O)Nc1ccc(Oc2ccnc3cc(OCCCN4CCCCC4)c(OC)cc23)c(F)c1)c1ccccc1 -10.00374639143023

7. COc1cc2c(Oc3ccc(NC(=O)NN=C(C)c4ccccc4)cc3F)ccnc2cc1OCCCN1CCCCC1 -10.116875164553585

8. COc1ccc(-n2nc(C(=O)Nc3ccc(Oc4ccnc5cc(OCCCN6CCCCC6)c(OC)cc45)c(F)c3)n(C)c2=O)cc1 -10.218632169786542

9. COc1cc2c(Oc3ccc(NC(=O)NN4C(=O)CSC4c4c(F)cccc4F)cc3F)ccnc2cc1OCCCN1CCCC1 -10.256847106153646

10. COc1cc2c(Oc3ccc(NC(=O)NS(=O)(=O)Cc4ccc(C)cc4)cc3F)ccnc2cc1OCCCN1CCOCC1 -10.27692109634116

11. COc1cc2c(Oc3ccc(NC(=O)NN=Cc4ccc(Cl)cc4)cc3F)ccnc2cc1OCCCN1CCC(C)CC1
    -10.341580836295224

12. COc1cc2c(Oc3ccc(NC(=O)/C=C/S(=O)(=O)c4ccccc4OC)cc3F)ccnc2cc1OCCCN1CCCCC1
    -10.440497903230485

13. COc1cc2c(Oc3ccc(NC(=O)c4nn(-c5ccccc5Cl)c5cc(F)ccc5c4=O)cc3F)ccnc2cc1OCCCN1CCC(C)CC1
    -10.468043614008312

14. COc1cc2c(Oc3ccc(NC(=O)NN=Cc4ccc(F)cc4)cc3F)ccnc2cc1OCCCN1CCCCC1
    -10.527305631989824

15. COc1cc2c(Oc3ccc(NC(=O)Nc4nc5c(C)ccc(C)c5s4)cc3F)ccnc2cc1OCCCN1CCN(C)CC1
    -10.559320239451706

16. COc1cc2c(Oc3ccc(NC(=O)NS(=O)(=O)Cc4ccccc4)cc3F)ccnc2cc1OCCCN1CCOCC1
    -10.629062637222779

17. COc1cc2c(Oc3ccc(NC(=O)c4nnn(-c5ccccc5C)c4C)cc3F)ccnc2cc1OCCCN1CCC(C)CC1
    -10.708130046611966

18. COc1cc2c(Oc3ccc(NC(=O)c4nnn(-c5ccccc5C)c4C)cc3F)ccnc2cc1OCCCN1CCCCC1
    -10.75201130373146

19. COc1cc2c(Oc3ccc(NC(=O)c4nn(-c5ccc(F)cc5)c(=O)n4C)cc3F)ccnc2cc1OCCCN1CCC(C)CC1
    -10.850928370666724

20. COc1cc2c(Oc3ccc(NC(=O)c4nn(-c5cc(C)ccc5C)c5ccccc5c4=O)cc3F)ccnc2cc1OCCCN1CCOCC1
    -10.90736400619354

21. COc1cc2c(Oc3ccc(NC(=O)c4cc(-c5ccccc5)ccn4)cc3F)ccnc2cc1OCCCN1CCCCC1
    -10.944002041495516

22. COc1cc2c(Oc3ccc(NC(=O)c4nn(-c5ccc(Cl)cc5Cl)c5ccccc5c4=O)cc3F)ccnc2cc1OCCCN1CCCCC1
    -11.010603179910028

23. COc1cc2c(Oc3ccc(NC(=O)c4nn(-c5ccc(C)cc5)c5ccccc5c4=O)cc3F)ccnc2cc1OCCCN1CCOCC1
    -11.077708041026122

24. COc1cc2c(Oc3ccc(NC(=O)c4cn(-c5ccccc5Cl)nn4)cc3F)ccnc2cc1OCCCN1CCCC1
    -11.14449718579243

25. COc1cc2c(Oc3ccc(NC(=O)NS(=O)(=O)Cc4ccc(F)cc4)cc3F)ccnc2cc1OCCCN1CCCCC1
    -11.229911129679115

26. COc1cc2c(Oc3ccc(NC(=O)c4c(Cl)c5ccccc5n(-c5ccc(F)cc5)c4=O)cc3F)ccnc2cc1OCCCN1CCOCC1
    -11.28329193813796

27. COc1cc2c(Oc3ccc(NC(=O)c4nnn(-c5ccccc5C(F)(F)F)c4C(F)(F)F)cc3F)ccnc2cc1OCCCN1CCN(C)CC1
    -11.3060688231394

28. COc1cc2c(Oc3ccc(NC(=O)c4nnn(-c5ccccc5C(F)(F)F)c4C)cc3F)ccnc2cc1OCCCN1CCOCC1
    -11.354432508931753

29. COc1cc2c(Oc3ccc(NC(=O)c4cc(-c5ccccc5)ccn4)cc3F)ccnc2cc1OCCCN1CCN(C)CC1
    -11.380181174324182

30. COc1cc2c(Oc3ccc(NC(=O)c4nn(-c5cccc(F)c5)c(=O)c5ccccc45)cc3F)ccnc2cc1OCCCN1CCCC1
    -11.407100628036758

31. COc1cc2c(Oc3ccc(NC(=O)c4nnn(-c5ccccc5C(F)(F)F)c4C(F)(F)F)cc3F)ccnc2cc1OCCCN1CCOCC1
    -11.449923572095255

32. COc1cc2c(Oc3ccc(NC(=O)c4nnn(-c5ccccc5C(F)(F)F)c4C(F)(F)F)cc3F)ccnc2cc1OCCCN1CCCC1
    -11.49608662779955

33. COc1cc2c(Oc3ccc(NC(=O)c4nn(-c5ccc(F)cc5)c5ccccc5c4=O)cc3F)ccnc2cc1OCCCN1CCC(C)CC1
    -11.546155255743054

34. COc1cc2c(Oc3ccc(NC(=O)c4nn(-c5cccc(C(F)(F)F)c5)c5ccccc5c4=O)cc3F)ccnc2cc1OCCCN1CCCCC1
    -11.600851900298952

35. COc1cc2c(Oc3ccc(NC(=O)c4c(C)n(-c5ccccc5Cl)c(=O)n4C)cc3F)ccnc2cc1OCCCN1CCN(C)CC1
    -11.640341597115352

36. COc1cc2c(Oc3ccc(NC(=O)c4nn(-c5ccccc5F)c5ccccc5c4=O)cc3F)ccnc2cc1OCCCN1CCC(C)CC1
   -11.705001337069415

37. COc1cc2c(Oc3ccc(NC(=O)c4nnn(-c5ccccc5)c4C)cc3F)ccnc2cc1OCCCN1CCOCC1
   -11.831464114782504

38. COc1cc2c(Oc3ccc(NC(=O)NS(=O)(=O)Cc4ccc(F)cc4)cc3F)ccnc2cc1OCCCN1CCCC1
   -11.860354039531494

39. COc1cc2c(Oc3ccc(NC(=O)c4nnn(-c5ccccc5)c4C)cc3F)ccnc2cc1OCCCN1CCCC1
   -11.99248313799697

40. COc1cc2c(Oc3ccc(NC(=O)c4cc(-c5ccc(C)cc5)ccn4)cc3F)ccnc2cc1OCCCN1CCN(C)CC1
   -12.07155054738616

41. COc1cc2c(Oc3ccc(NC(=O)c4nc5ccccc5n(-c5ccccc5Cl)c4=O)cc3F)ccnc2cc1OCCCN1CCN(C)CC1
   -12.162827172829553

42. COc1cc2c(Oc3ccc(NC(=O)c4nc5ccccc5n(-c5ccc(F)cc5)c4=O)cc3F)ccnc2cc1OCCCN1CCN(C)CC1
   -12.333171207662138

We can immediately observe that all ligands are relatively similar in structure and arrangement, and that the experimental affinity gradually (an unconditionally) improves as the chain progresses. Figure 5 shows what a modifications from one molecule in the chain to the next looks like, showing the first three ligands from this chain. We can very clearly see the minimal, single change that occurs at every step. From ligand 1 to ligand 2, the terminal heterocycle on the far right changes from containing one oxygen and one nitrogen to containing two nitrogens and a methyl group. From ligand 2 to ligand 3, the leftmost group is dropped and additional fluorines are added. It is crucial that changes are minimal and realistic, as we want an LLM to be able to rationalize any of the changes in a few interpretable steps.

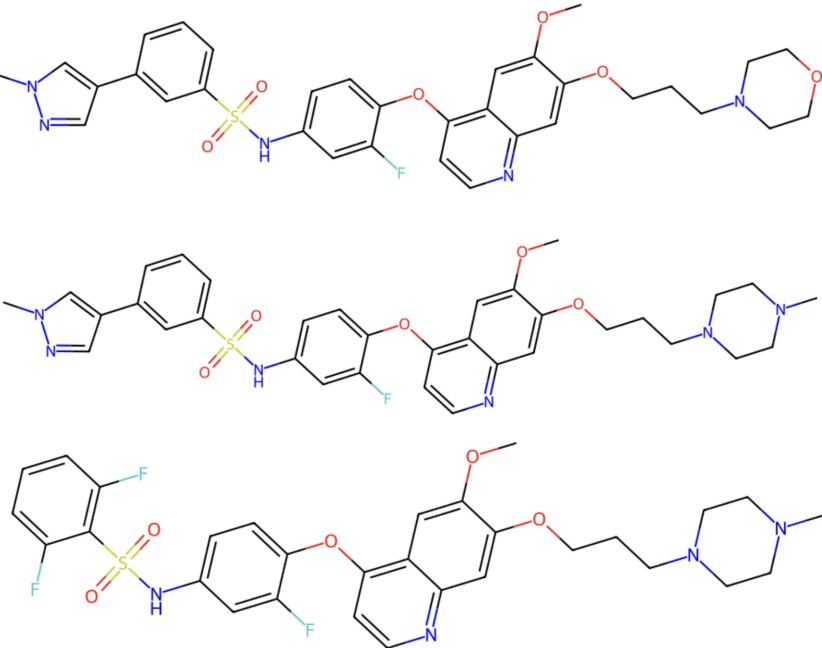

Figure 5: Visualizations of the first 3 ligands in this example chain. Observe how modifications are gradual and resemble changes that might derive from a legitimate reasoning process.

Figure 6 further demonstrates this by showing the final two ligands from this chain. The similarities to the initial ligands are clear, and one can imagine how the gradual steps along the chain may have led to these significant changes. Crucially, even here, we can observe minimal changes being made.

Figure 6: Visualizations of the last 2 ligands in this example chain. Notice the obvious similarities to the start of the chain, and notice how changes are still minimal and gradual here.

According to the affinities from the chain, these two ligands have experimental affinities of over -12 kcal/mol, while the starting molecules had affinities hovering around -8 kcal/mol. From this, we can directly visualize how these ligand chains might guide an LLM toward more informed modifications of ligands; each change is gradual and reasonable, yet at the same time gradually progressing the desired binding affinity.

## C.2  LLM Prompts For Dataset Formation

This section provides the exact prompts used to create the supervised fine-tuning dataset used in this work.

Consider one of the ligand chains formed by the clustering-sorting process. For each ligand/position in the chain, we first ask the LLM to generate a summary based on all the past (weaker affinity) ligands in the chain. This summary is used in the input for SFT, simulating the information the LLM might receive for an optimization step.

---

**Prompt to generate the summary of past ligand modifications used in the input for SFT**

You are a chemistry-aware assistant that is collaborating with me on generating a ligand for a protein with high binding affinity. Below is a chronological history of past ligands you've generated. Provide a summary of changes and modifications you've made so far in regards to the ligand structure and how it impacts the binding affinity; the goal is to give context about past iterations to another agent. Be sure to explicitly output the SMILES of every past ligand. Do not provide any suggestions for future generations at this time. Keep your response relatively short.
*SMILES: Affinity*
*SMILES: Affinity*
*SMILES: Affinity*
...

---

Where we input all previous ligands and their binding affinities in the chain as *SMILES: Affinity*. The generated summary is placed into the following format, which becomes the full input for SFT:

---

**Full SFT Input**

We are collaborating on generating a ligand for a protein with high binding affinity. I will give you the output from docking software after each of your attempts. Provided below is a brief summary of past ligand modifications:
****GENERATED SUMMARY****
First describe what you have learned from the above summary. Then based on that knowledge, generate a ligand that can bind to this protein with high binding affinity. Ensure that your generation is unique and is not found within the provided data. Follow this format for your final answer: \\box{MOLECULE}, where MOLECULE is your proposed ligand in SMILES format.
*SMILES: Affinity*
*SMILES: Affinity*
*SMILES: Affinity*
...

---

After this, we ask the LLM to generate reasoning that might lead an agent to generate the next (stronger affinity) ligand in the chain. This becomes the full desired output for SFT. In essence, we want to tune the model to carry out what we know to be a strong modification decision for a particular protein target based off information provided in the input.

---

**Prompt to generate the full output for SFT:**

We are collaborating on generating a ligand for a protein with high binding affinity. I will give you the output from docking software after each of your attempts. Provided below is a brief summary of past ligand modifications:
****GENERATED SUMMARY****
Based on the summary, first pretend that you are thinking about what modifications to make to create a stronger molecule. In the end, choose to generate the molecule *NEXT_LIGAND*. Do not mention this molecule until you've explained what decisions and specific structural modifications would lead to this new molecule. It's crucial that you pretend to have came up with this generation yourself after some brief analysis of the provided summary. Provide the final ligand explicitly in this exact format, character by character: \\box{*NEXT_LIGAND*}. Keep your response relatively short.
*SMILES: Affinity*
*SMILES: Affinity*
*SMILES: Affinity*
...

---

Notice that the first half of this prompt is the same as the actual input for SFT. However, we instead tell the LLM to pretend like it's doing the generation action, using the predetermined *NEXT_LIGAND* as the guide for its explanation, where *NEXT_LIGAND* is the next (higher affinity) ligand in the ligand chain. This results in an SFT output that not only contains the desired next ligand, but also provides a sample explanation for how you might modify previous ligands to create the new one. This is precisely what we want for the SFT dataset, as we want a model to learn how to make the best modifications for any given input to optimize toward a point of higher affinity. Here is a sample LLM output for this prompt:

---

**Sample Full SFT Output**

Considering the previous ligand's balance of hydrophobic aromatic rings, heterocycles, and hydrogen-bonding groups, I think introducing a more rigid and cyclic structure could enhance binding affinity. Replacing flexible linker regions with nitrogen-containing heterocycles might improve interactions through additional hydrogen bonds and electrostatic contacts. Additionally, modifying substituents to include carbonyl groups could further strengthen hydrogen bonding capacity. Incorporating a carbamate moiety at a strategic position can also influence binding strength. After analyzing these features, I realize that a cyclic structure incorporating these elements—specifically, a fused ring system with carbamate and aromatic groups—optimize interactions with the protein binding site. Therefore, I plan to generate a ligand with these features, resulting in the target molecule: \\box{*NEXT_LIGAND*}.

---

### C.3 SFT TRAINING PROCESS

In this section, we provide information about the training process for our supervised fine-tuned model. We fine-tune using the unified open-source fine-tuning repository LLaMA-Factory Zheng et al. (2024). We utilize Low-Rank Adaption (LoRA) Hu et al. (2021) to train a subset of the model parameters, saving a significant amount of time and computation. We utilize all default hyperparameters from the LLaMA-Factory repository (see the llama3_lora_sft.yaml example file in examples/train_lora/), except for modifying the train-validation split to be 0.95/0.05 instead of 0.90/0.10.

Figure 7 provides the training and validation loss graphs for this process for the BRD4 target. We train for 10 epochs on a dataset of around 5,000 samples, taking around 150 minutes on a NVIDIA H200. As is evident from the figure, validation loss drops rapidly, then rather quickly plateaus, and increases rapidly as the model overfits to the relatively small dataset. We let the training continue past the overfitting point just for the chance of any emergent behavior. However, we carefully select the checkpoint for which the validation loss is at its minimum, which we evaluate to be the checkpoint at step 2,000. We merge the LoRA adapters at this checkpoint into the original base model to obtain the fine-tuned model used in MOLLEO optimization.

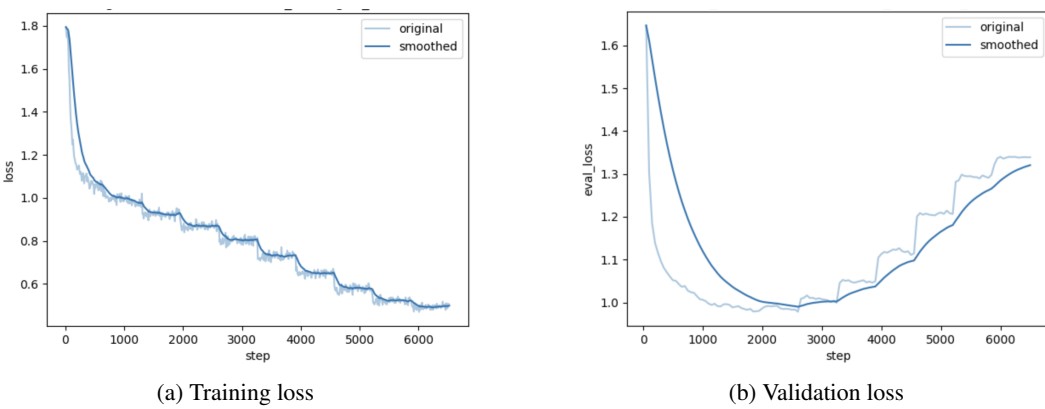

(a) Training loss          (b) Validation loss

Figure 7: Training and validation loss graphs for supervised fine-tuning

### C.4 FURTHER COMPARISON OF TUNED VS UNTUNED LLAMA

In this section, we further compare the results of our fine-tuned Llama model with the untuned model. Specifically, we account for the high amount of invalid format responses that the untuned model yields in order to provide a more focused comparison of the generated molecules themselves.

We do this by only considering the first $n$ LLM generations of the fine-tuned model, where $n$ is the total number of validly-formatted responses made by the untuned model throughout the entire 1000-step process. In this way, we take the generation results from an equal number of valid LLM attempts from both models, allowing us to purely compare the quality of molecule generations between the models without statistical interference from answers with invalid formats. The results from this limitation are shown in Table 6:

Table 6: Boltz-2 affinity (kcal/mol) for untuned Llama vs fine-tuned Llama w/ limitation applied

| Method | Mean $\pm$ SD (c-MET) | Mean $\pm$ SD (BRD4) |
|---|---|---|
| Untuned Llama | -10.8 $\pm$ 0.4 | -10.7 $\pm$ 0.7 |
| Fine-tuned Llama w/ limitation | **-11.1** $\pm$ 0.3 | **-10.9** $\pm$ 0.3 |

With this limitation, the fine-tuned model yields $p = 0.02$ for c-MET and $p = 0.15$ for BRD4. However, we can consider a joint probability for the hypothesis that the fine-tuning method improves generations across all targets, since the runs are entirely independent. Using Fisher's method with 4 degrees of freedom, we get a joint probability of $p_{combined} \approx 0.02$, giving us statistically significant

evidence that our fine-tuning framework improves the inherent generation quality of the LLM for any protein target $p$.

This limitation is actually heavily biased toward the untuned model, since its model generations are built upon existing strong molecules generated by default crossover/mutation operators throughout the entire course of the optimization process, while the fine-tuned results are limited only to the initial parts of the optimization process. Even with such a limiting ablation, we observe a statistically significant increase in binding affinity with the fine-tuned model, concretely supporting the claim that our post-training framework improves the inherent quality of molecule generations within the optimization process.

## D    DEMONSTRATION FOR LACK OF BINDINGDB LIGANDS

In this section, we demonstrate that for our supervised fine-tuning dataset, we have a workaround in the situation where we are optimizing for a protein target that is not well studied and has few results for experimentally-tested ligand binders.

Our original dataset for c-MET was sufficiently large because c-MET is a very well-studied target. As a proof of concept however, we pretended that this was not a well-studied target, and instead formed our dataset by considering 20 protein targets that we determined to have structural similarities c-MET using the BLASTP tool Altschul et al. (1990). We took the ligand entries corresponding to these 20 targets and applied the same process to form a surrogate dataset. We trained the same small Llama model on this dataset, and its performance in MOLLEO is shown in Table 7.

Table 7: Boltz-2 affinity (kcal/mol) for Llama tuned on 2.5k vs 30k datasets for c-MET

| Method | Mean $\pm$ SD |
|---|---|
| Llama | -10.8 $\pm$ 0.4 |
| Llama (c-MET dataset) | -11.7 $\pm$ 0.1 |
| Llama (surrogate dataset) | -11.6 $\pm$ 0.2 |

Comparing the results we get from a model trained on the pure c-MET dataset against our surrogate dataset, we get $p = 0.13$ from a two-sided independent Student's t-test, which implies a non-significant difference. We see that if the desired protein target does not have sufficient ligand entries, we can make up for it by identifying protein targets that are structurally similar to it and use their ligand entries instead. This guarantees some level of similarity in the input ligands, and as demonstrated experimentally, does not hurt performance relative to using a dataset comprised only of target-specific ligands.

