# OpenReview forum: "MOLLEO+: Towards Optimized Use of LLMs for Drug Discovery"
_ICLR.cc/2026/Conference — Submitted to ICLR 2026_

### Official Review · Reviewer_1zdh · 2025-10-27

**Soundness:** 2
**Presentation:** 2
**Contribution:** 2
**Rating:** 4
**Confidence:** 3

**Summary:**

This paper proposed MOLLEO+, an optimized LLM workflow that improves MOLLEO for de novo molecule generation. Specifically, they replace docking with the recently released biomolecular foundation model Boltz-2 as an oracle which can improve the predicted binding affinity. Additionally, they incorporate knowledge of existing ligands and propose a fine-tuning strategy to better modify existing ligands towards higher activity. Those components together demonstrate the superiority of MOLLEO+ on the receptor tyrosine kinase c-MET and the BRD4 protein.

**Strengths:**

- Replace the docking-based fitness evaluator with Boltz-2, which increases the mean Absolute Binding Free Energy of generated molecules by over 100%.
- Utilize a starting population of ligands based on BindingDB, which increases mean predicted binding affinity of generated compounds by up to 15%
- Propose a novel post-training framework to fine-tune LLMs for lead optimization tasks using a semi-synthetic dataset, which can improve  the quality of its generated molecules

**Weaknesses:**

- The experiments only evaluate two targets, c-MET and the BRD4. It’s unclear that if this method can be applicable to more broad targets.
- Using Boltz-2 to evaluate ABFE also introduces bias by the computational method itself.
- This method may be less effective for those targets that lack sufficient ligand data.
- Generated molecules have less novelty due to the use of strong known binders from BindingDB as the starting population

**Questions:**

Please refer to the weakness section

---

> ### Author Response · Authors · 2025-12-03
> **Rebuttal by Authors**
>
> We thank the reviewer for their valuable feedback and insights about our work.
>
> >**Q1**: The experiments only evaluate two targets, c-MET and the BRD4. It’s unclear that if this method can be applicable to more broad targets.
>
> **A1**: We understand your concern about this limitation. However, since c-MET and BRD4 are vastly different proteins in terms of structure and activity (receptor tyrosine kinase vs bromodomain-containing protein), we feel that the consistent results across both targets demonstrate a level of robustness supported simply by the diverse protein space covered by c-MET and BRD4. Because of this, generalizability to other targets would not be difficult within this method, particularly to other targets within the same families of proteins as c-MET and BRD4 (e.g. other tyrosine kinases), which include many proteins that the field is often most interested in. In a practical standview, we are likely to generalize to other useful targets, by virtue of the relatively diverse and vast space that c-MET and BRD4 jointly cover. We are also constrained by the computational cost of expanding to other targets, and observe that several other established works similarly only evaluate two primary targets [1].
>
> [1] Peter Eckmann, Dongxia Wu, Germano Heinzelmann, Michael K. Gilson, and Rose Yu. MF-LAL: Drug compound generation using multi-fidelity latent space active learning, 2025. URL https://arxiv.org/abs/2410.11226.
>
> >**Q2**: Using Boltz-2 to evaluate ABFE also introduces bias by the computational method itself.
>
> **A2**: There may be a misunderstanding here. Boltz-2 is not directly used to "evaluate ABFE", instead we establish a strong correlation between Boltz-2 and ABFE, from which we can make deductions about the relative improvements in the cheaper Boltz-2 oracle. While we acknowledge that Boltz-2 has computational biases related to its training set, these biases are not particularly related to the argument that we are trying to make. The observed correlation has strong implications about the usefulness of Boltz-2 in consistently acquiring higher ABFE results, which has no relation to the built-in computational biases of Boltz-2 itself. We apologize if we have misunderstood your concern, and would appreciate any clarification if that is the case.
>
> >**Q3**: This method may be less effective for those targets that lack sufficient ligand data.
>
> **A3**: We understand your concern about this limitation. To address this problem, we include a brief analysis in Appendix D that shows how might work around this problem in the case of evaluating a target with insufficient ligand data. We demonstrate that for the fine-tuning problem, forming a larger dataset with ligands from structurally-adjacent protein targets allows us to achieve essentially the same performance, meaning that in the case of insufficient ligand data, we have an alternative solution. For the BindingDB starting population in MOLLEO+, we acknowledge that this might be a realistic limitation; however, we observe that using MOLLEO with Boltz-2 and ZINC 250K already yields a significant gain from the original MOLLEO algorithm, and thus we implicitly offer an alternative solution in this case as well. Ultimately however, we would like to state that our method is inherently intended for lead-optimization tasks where the target is already relatively well studied.
>
> >**Q4**: Generated molecules have less novelty due to the use of strong known binders from BindingDB as the starting population
>
> **A4**: We acknowledge this concern and possible limitation. We have updated section 4.3 in the manuscript to include further analysis of novelty, including two different similarity thresholds and a filter employing Murcko scaffold-level similarity. These results show that although novelty from MOLLEO+ is lower on average than MOLLEO, we still find improved results within a restricted subset of novel molecules. As stated within the paper analysis, we acknowledge that MOLLEO+ does suffer from decreased novelty overall. However, we do not believe this necessarily hinders the practical utility of the method, and argue that for our intended points of optimization (translation to real-world practicality), the novelty results we have shown with several filters are sufficient to justify its usefulness in scenarios such as lead optimization and beyond. Ultimately, we are looking for molecules that are very strong binders that are perhaps dissimilar enough to be considered novel; we don't believe that a hard threshold much tighter than Tanimoto < 0.4 is justified or necessary for many practical applications. Thus we are comfortable in claiming that MOLLEO+ generations are sufficiently novel enough for the primary purpose that we are trying to achieve.
>
> We thank the reviewer for their time and consideration in this process.

---

### Official Review · Reviewer_gqGE · 2025-10-29

**Soundness:** 1
**Presentation:** 2
**Contribution:** 1
**Rating:** 0
**Confidence:** 4

**Summary:**

The authors propose MOLLEO+, an extension to the existing MOLLEO framework for molecular optimization via the use of large language models. The authors demonstrate that the use of a Boltz-2 oracle and an improved starting population leads to better generated molecules (in terms of targeted molecular properties). The authors also suggest a supervised fine-tuning framework for molecular editing, but show that the fine-tuned model still underperforms larger models. In my opinion, the work that the authors propose here falls within the way that the original MOLLEO was designed to be used. Critically, their work does not extend the capabilities of the original framework. While their supervised fine-tuning approach looks promising, it is underevaluated. Thus, this work is not worthy of acceptance to ICLR.

**Strengths:**

The authors show the intended use of MOLLEO in this work and the benefits that it can lead to. This is instructive for users who are curious about adopting MOLLEO in their molecular design efforts.

**Weaknesses:**

Naturally, it should be trivial that when using MOLLEO, one chooses an oracle that is supposed to correlate to the property that they care about (the authors choose ABFE). I don’t think it’s a surprising result that Boltz-2 gives better correlations for ABFE as it was trained to reproduce binding affinities. Since docking is constructed to be extremely approximate and is designed for speed in screening extremely large libraries, I also don’t think it’s surprising that using a docking oracle results in poor estimates for ABFE. In light of this, Section 4.1 appears to be a very limited benchmark of oracle performance confounded by MOLLEO’s evolutionary search, since MOLLEO + AutoDock is attempting to maximize properties (i.e docking scores) completely uncorrelated to their intended target. An independent benchmark on a set of molecules independent of MOLLEO’s structural generation would have been much more appropriate.

The authors also report that the molecules have better Boltz-2 affinity scores because they use a better starting population correlated to their property of interest. In the original MOLLEO work, the authors have already evaluated something equivalent (best molecules in ZINC 250K) in Table 3. Again, the increased performance of MOLLEO+ versus the unmodified MOLLEO (Boltz-2) comes from aligning the starting population towards their intended goal, which seems to me as the normal way to use MOLLEO to begin with. MOLLEO was never designed to be strictly used only with ZINC 250K, and so it is confusing to me that the authors frame the correct usage of MOLLEO for their problem as an optimization of the original framework.

The authors also do not report (or maybe I missed it somewhere) if the results of their runs for each method were the result of multiple runs with random seeds, so it’s hard for me to assess if the performance of their method is consistent.

The supervised fine-tuning method shows promise -- it would indeed be desirable if the molecular edits by the LLM encapsulate the way that medicinal chemists would think. However, Tanimoto similarity is not a smooth metric, given that closely related structures (i.e changing functionalizations on a backbone/template) can also result in large differences in Tanimoto similarity. While it is also the cornerstone of structure-activity relationships, there’s also no guarantee that the same structural edits necessarily correlate with activity across different backbones or chemical environments. Without being able to see what these “ligand chains” look like, it’s hard to rationalize with physicochemical principles that the models are being fine-tuned on the correct concepts. The aggressiveness of the fine-tuning also is not evaluated, as one can imagine that in a strongly overfit case, all starting structures immediately collapse towards their respective cluster’s local maxima in the training data. In a real-world case, accessing a dataset that is large enough for a model to learn these edits relevant for their target protein may not exist as well, so it would have been interesting to see if there could be some level of general chemistry knowledge learned from a dataset not directly related to their optimization task.

**Questions:**

1) Were there multiple runs for each method with a random seed?
2) Was your BindingDB initial population sorted by experimental or Boltz-2 binding affinity?
3) What is an example of this ligand chain? Do you observe these chain edits being used over the course of the MOLLEO+ run? Do these chain edits actually result in a positive chain in molecular property?
4) How do you support the claim that “GPT-4.1mini is known to be a stronger model overall”? Is there a relevant source?

---

> ### Author Response · Authors · 2025-12-03
> **Rebuttal by Authors**
>
> We thank the reviewer for their valuable feedback and insights about our work.
>
> >**Q1**: Were there multiple runs for each method with a random seed?
>
> **A1**: Yes there were, we have updated the manuscript (section 4.2) to include this information. Each method was run 3 separate times with 3 different random seeds. We report the standard deviation across runs for all crucial metrics (binding affinity) to affirm rigor and consistency between runs.
>
> >**Q2**: Was your BindingDB initial population sorted by experimental or Boltz-2 binding affinity?
>
> **A2**: Our BindingDB initial population is sorted and selected by experimental affinity. This guarantees that the structures within the starting population used by the algorithm are experimentally-verified to be strong, which Boltz-2 can then optimize for, given the strong demonstrated correlations between Boltz-2 and the most accurate free energy calculation methods from section 4.1.
>
> >**Q3**: What is an example of this ligand chain? Do you observe these chain edits being used over the course of the MOLLEO+ run? Do these chain edits actually result in a positive chain in molecular property?
>
> **A3**: We have updated the manuscript (Appendix C.2) to include an example of a ligand chain, along with visualizations of certain molecules. We observe that the chain modifications are gradual and reasonable, and in terms of experimental binding affinity, these edits do - by design - result in positive molecular changes over the course of the chain. It is difficult to evaluate whether or not these edits are directly "used" across a MOLLEO+ run; we sample from our LLM with top-p = 0.75 and top-k = 50, resulting in a lot of natural variance. However, we can infer from the drastically improved results over the base model (Table 2, Table 3) that the LLM learns from the samples given in the chains to make more guided decisions as a whole.
>
> >**Q4**: How do you support the claim that “GPT-4.1mini is known to be a stronger model overall”? Is there a relevant source?
>
> Please refer to **A7** under rebuttals for Reviewer UiFu.
>
> >**Q5**: I don’t think it’s a surprising result that Boltz-2 gives better correlations for ABFE as it was trained to reproduce binding affinities. Since docking is constructed to be extremely approximate and is designed for speed in screening extremely large libraries, I also don’t think it’s surprising that using a docking oracle results in poor estimates for ABFE.
>
> **A5**: We respectfully disagree. We argue that it is not an obvious result that Boltz-2 gives better correlations for ABFE. Boltz-2 was trained to reproduce experimental binding affinities from datasets such as BindingDB, not to reproduce expensive physics-based, free-energy calculations like ABFE. The fact that it has a strong correlation among generated, out-of-distribution molecules was not obvious to us, and reflects a relationship in predictive accuracy that we don't believe it was explicitly trained for. And while it's true that docking was constructed to be most effective in large-scale screenings, nearly all current studies still utilize docking for evaluating the binding affinity quality of their generative approaches. Our aim in section 4.1 was to show that Boltz-2 can be used in place of the current industry-standard for evaluation (docking) and yield a drastic increase in ABFE affinity, something that is not entirely obvious given that both methods provide binding affinity predictions, and especially considering that docking affinity predictions are often (incorrectly) taken as an accurate measure in relevant studies. Ultimately, both methods have been trained to provide a predictive measure of binding affinity, and to show that Boltz-2 can actually produce higher ABFE scores in practice within a real generative framework is a result that - to our knowledge - has not been demonstrated in other works.
>
> >**Q6**: Again, the increased performance of MOLLEO+ versus the unmodified MOLLEO (Boltz-2) comes from aligning the starting population towards their intended goal, which seems to me as the normal way to use MOLLEO to begin with.
>
> **A6**: We understand and acknowledge this concern; however, we argue that the interesting point about this change comes from the fact that this is a starting population chosen and evaluated by real-world experimental binding affinities. The purpose of our framework is to improve the capabilities of MOLLEO for application to real-world drug discovery. Using experimental data directly from BindingDB within this framework introduces another layer of real-world practicality that we believe the original MOLLEO algorithm crucially lacked. We don't believe that the results of this change are trivial; aligning the algorithm toward a real-world focus and observing tangible improvements in metrics demonstrates a shift in how the algorithm can be used in a practical drug-discovery environment.
>
> We thank the reviewer for their time and consideration in this process.

---

### Official Review · Reviewer_MezY · 2025-10-31

**Soundness:** 2
**Presentation:** 3
**Contribution:** 2
**Rating:** 4
**Confidence:** 3

**Summary:**

The authors present Molleo+, an iteration of the original Molleo method, in which the proxy for the binding energy oracle is replaced with Botz2, a different initial set of molecules is used, and an additional fine-tuning step is introduced on paired low- and high-affinity binders to better focus the model on optimizing affinity. The approach is evaluated on the task of identifying higher-affinity binders for two target proteins, c-MET and BRD4.

**Strengths:**

1.	The problem statement and main contributions are clearly articulated, and the manuscript is easy to follow. The task of improving binding affinity is important in the life sciences, and the proposed approach is well-suited to addressing it.
2.	Molleo with Boltz2 and Molleo+ demonstrate strong performance improvements across the reported metrics, in some cases making the Molleo lineage competitive with or superior to other baselines.
3.	The authors also provide useful insights into the factors driving these results, such as the correlation analysis presented in Figure 3

**Weaknesses:**

The central theme is that the evaluation requires greater robustness:

1.	Demonstrating meaningful benefit: More evidence is needed to show that the Molleo+ approach provides a genuine performance improvement (Table 2 and Table 3). First, the overrepresentation of high-affinity binders in the training data may be driving the observed results (see Question 1). Second, since binding affinity is evaluated using Boltz2, Molleo+ may effectively be distilling Boltz2 rather than making genuinely improved affinity predictions (see Question 2).

2.	Target-dependent performance: There is also a risk that the reported gains are partly due to the particularly poor Autodock binding scores for the selected targets (as illustrated in Figure 3). Consequently, the observed improvement may not generalize to other targets. (see Question 3)

Without the above the novelty would be limited to using higher quality data (via Boltz2) as input to the original Molleo.

**Questions:**

1.	Filtered Mean analysis (Table 3): The Filtered Mean is computed based on ligands with scores below a threshold of 0.5. What is the resulting average maximum similarity for this subset? If it exceeds 0.35 for c-MET and BRD4 (the value observed for Molleo and Molleo (Boltz2)), the authors could examine a subset of data with a mean similarity of 0.35. This would enable a fairer comparison with Molleo (Boltz2) and Molleo.

2.	Orthogonal evaluation of binding affinity: For Molleo+, it would strengthen the work to include an orthogonal method for estimating binding affinity. Specifically, could the evaluation using ABFE values from Table 1 also be performed for BRD4, and for Molleo+ across both targets?

3.	Effect of affinity quality: The authors could further analyze performance at different levels of correlation to determine whether there is a direct relationship between the quality of affinity estimates and predictive performance. For instance, controlled noise could be added to the c-MET binders to progressively reduce correlation and assess sensitivity.

---

> ### Author Response · Authors · 2025-12-03
> **Rebuttal by Authors**
>
> We thank the reviewer for their valuable feedback and insights about our work.
>
> >**Q1**: Filtered Mean analysis (Table 3): The Filtered Mean is computed based on ligands with scores below a threshold of 0.5. What is the resulting average maximum similarity for this subset? If it exceeds 0.35 for c-MET and BRD4 (the value observed for Molleo and Molleo (Boltz2)), the authors could examine a subset of data with a mean similarity of 0.35. This would enable a fairer comparison with Molleo (Boltz2) and Molleo.
>
> **A1**: We believe our current comparison to be quite fair; the resulting average maximum similarity for MOLLEO+ under this filter is 0.39, which we believe to be a very minimal increase over the mean for the other two methods. However, for your consideration, we recalibrated this filter to only accept molecules under Tanimoto similarity = 0.45, which readjusts the mean of the MOLLEO+ subset to be around 0.36. You can find the resulting affinity comparisons in this case in Table 4 in section 4, under Sim = 0.45. Even at this tighter constraint and more balanced mean similarity, MOLLEO+ still achieves p<0.01 compared to the best baseline. Further updated similarity and novelty analysis and discussion is also included in this section for your consideration. These results support the fact that the overrepresentation of high-affinity binders in the BindingDB starting population are not driving the results in a biased way, as we still find stronger results within subsets of guaranteed novelty.
>
> >**Q2**: Orthogonal evaluation of binding affinity: For Molleo+, it would strengthen the work to include an orthogonal method for estimating binding affinity. Specifically, could the evaluation using ABFE values from Table 1 also be performed for BRD4, and for Molleo+ across both targets?
>
> **A2**: Due to the vast computational demands of obtaining ABFE results, we are unfortunately unable to provide these requested experiments at this time. However, we support the strength of our results through the established correlation between Boltz-2 and ABFE shown in section 4.1. We argue that these additional results are not a necessity for our claim, since the strong correlation shown in this section supports that a large relative increase in Boltz-2 is close to a “ground-truth” increase in binding affinity. This is why we establish this correlation in this section, as large-scale ABFE results are extremely expensive and often unreasonable to obtain.
>
> >**Q3**: Effect of affinity quality: The authors could further analyze performance at different levels of correlation to determine whether there is a direct relationship between the quality of affinity estimates and predictive performance. For instance, controlled noise could be added to the c-MET binders to progressively reduce correlation and assess sensitivity.
>
> **A3**: We are not entirely clear on what the reviewer is suggesting in this particular comment, and would appreciate any clarification. If it is suggesting that we add controlled noise to the Boltz-2 fitness function within the MOLLEO algorithm, we don't necessarily see the value in this experiment. Since the genetic algorithm necessarily selects the top fitness-evaluated ligands for repopulation, adding noise to the fitness function would confound the process in a way that we don't think would show anything meaningful about generalizability or the quality of affinity estimates. As it relates to the concern raised in Weakness #2, we show in section 4.1 that docking is inaccurate for both c-MET and BRD4, while Boltz-2 does quite well for both. These are two very different targets and both structure and activity (receptor tyrosine kinase vs bromodomain-containing protein). We feel that this is strong enough to show that the gain in using Boltz-2 over AutoDock would be consistent across other targets, deriving from the observed consistency across the diverse protein space covered by c-MET and BRD4.
>
> We thank the reviewer for their time and consideration in this process.

---

### Official Review · Reviewer_UiFu · 2025-10-31

**Soundness:** 2
**Presentation:** 3
**Contribution:** 2
**Rating:** 4
**Confidence:** 3

**Summary:**

The paper presents an enhanced version of the generative lead-optimization framework MOLLEO, replacing the traditional docking oracle with the AI foundation model Boltz-2, seeding the starting population from BindingDB instead of ZINC to bias toward known actives, and fine-tuning a compact Llama model on a semi-synthetic dataset to improve crossover and mutation operations. Empirical validation focuses primarily on the target c MET (with ABFE comparisons) and a second target BRD4 (via Boltz-2 scoring) and claims that the proposed method outperforms the original docking-based baseline in terms of predicted and simulated binding affinities while maintaining medicinal chemistry viability.

**Strengths:**

1. The motivation to replace docking with a modern AI oracle is good and aligns with recent advances in binding-affinity modelling. The authors present data showing that Boltz-2 better correlates with ABFE on c MET than docking.
2. The switch to a BindingDB-derived starting pool is sensible: it reflects realistic chemical spaces of known actives and helps the algorithm exploit known chemotypes while still maintaining diversity. The clustering and fingerprint details are clearly described.
3. The semi-synthetic SFT for a Llama model is well described with prompt templates, training curves and validation splits. The ablation shows benefit of tuning the LLM operator.
4. The ABFE implementation is described in greater detail than is often seen: the authors provide schedule detail, pose preparation with alignment, and run time data which enhances reproducibility.
5. The authors include medicinal-chemistry relevant metrics and apply novelty filtering against the starting pool, which demonstrates awareness of downstream requirements beyond raw binding affinity.

**Weaknesses:**

1. External validity is limited: the strongest evidence comes from one target (c MET) for ABFE validation. For BRD4, only oracle scores (Boltz-2) are shown. That raises concerns about generalisability across target families.
2. The ABFE protocol lacks sufficient detail on convergence diagnostics (e.g., number of replicas, uncertainty estimates, window closures) and the starting-pose difference between arms (docking vs Boltz-2) may bias the comparison.
3. Baselines are not fully aligned: the change in oracle, starting pool and LLM are all conflated. There is insufficient control to isolate each contribution (oracle swap, starting pool shift, LLM fine-tuning) and the claim of GPT-4.1-mini being stronger than GPT-4 is not substantiated.
4. Novelty and chemical novelty analysis is limited: the paper uses a Tanimoto similarity cutoff against the starting pool but does not report scaffold novelty, nor similarities against broader medicinal-chemistry databases like ChEMBL or patent literature.
5. Statistical reporting is incomplete: confidence intervals and error bars are missing. Effect sizes and sample sizes vary across arms (e.g., 10 vs 20 samples) which complicates interpretation of statistical significance.
6. The synthetic SFT dataset raises quality questions: chemical validity and label fidelity of synthetic-data edits are not clearly verified (e.g., no PAINS filtering, no human spot checks) and the relative contribution of SFT vs other changes (oracle, seed pool) remains ambiguous.
7. Crucially, the choice of Boltz-2 as the oracle is not justified with full benchmarking against experimental binding affinity data on the chosen targets. Without that, the practical usefulness of the gains is unclear: if the correlation between Boltz-2 and experiment is low then improvements in the generative workflow may not translate into real-world value.
8. Similarly, the practical utility of metrics like QED and SA score are in question. Those are frequently referenced in scientific literature, however, it is a common knowledge that these metrics are poor proxies for real-world drug-likeness and synthetic accessibility.
9. The discussion section lacks deeper reflection on failure modes, target limitations, and trade-offs (e.g., cost vs quality, when to use this method vs classic workflows).

**Questions:**

1. On oracle choice and experimental correlation. What is the correlation between Boltz-2 predicted affinities and experimental binding measurements on c MET, BRD4 and ideally one more independent target family? Please report Pearson and Spearman coefficients with confidence intervals, calibration plots, and address any systematic bias across affinity ranges. How do these correlations compare with docking scores (AutoDock Vina or similar) and ABFE estimates? Given that Boltz-2 is cited as “near-FEP accuracy” in benchmarks, please provide specifics on this for the chemical space relevant to your study.
2. If the correlation to real experimental binding affinity is moderate, what are the practical implications of your workflow for medicinal-chemistry decision-making (hit-to-lead, lead optimisation)? What threshold would you consider sufficient for this method to be actionable in a drug-discovery funnel, and in which scenarios (library size, novelty requirement, target difficulty) would this workflow add meaningful value?
3. On ABFE protocol. Could you provide full details on how many independent replicas per ligand were run, the resulting uncertainties (e.g., standard error, confidence intervals) and convergence diagnostics (e.g., window overlap checks, cycle closure if any)? Also clarify whether both arms (docking-oracle vs Boltz-2) used identical simulation schedules, starting poses, restraints and stopping criteria, and provide the wall-clock time per ligand. If starting poses differ, then how do you mitigate pose-biasing the ABFE comparison?
4. On SFT dataset quality. How did you verify chemical validity and label fidelity of the semi-synthetic dataset? Did you apply automated checks (valence rules, PAINS filters, structural alerts) or human spot-checks? What fraction of generated edits did you discard for implausibility? Please provide ablation results isolating the effect of SFT only (keeping starting pool and oracle fixed) and report whether the fine-tuned LLM or the dataset quality was the limiting factor.
5. On novelty and chemical novelty. Could you expand novelty analysis to scaffold-level novelty, and maximum similarity to any known binder across BindingDB or ChEMBL? Also please show sensitivity of results to different similarity thresholds rather than a single cutoff. Provide counts of unique scaffolds generated and compare to baseline.
6. On baselines and fairness. Please include control runs that isolate each change independently: (a) original LLM + docking oracle + ZINC start, (b) original LLM + Boltz-2 oracle + ZINC start, (c) original LLM + docking oracle + BindingDB start, (d) fine-tuned LLM + docking oracle + BindingDB start, all under equal compute budget and same number of generations. Also consider including a small ABFE-guided genetic algorithm (without an LLM) to establish a cost-quality upper bound.

---

> ### Author Response · Authors · 2025-12-03
> **Rebuttal by Authors (Part 1)**
>
> We thank the reviewer for their valuable feedback and insights about our work.
>
> >**Q1**: On oracle choice and experimental correlation. What is the correlation between Boltz-2 predicted affinities and experimental binding measurements on c MET, BRD4 and ideally one more independent target family? Please report Pearson and Spearman coefficients with confidence intervals, calibration plots, and address any systematic bias across affinity ranges. How do these correlations compare with docking scores (AutoDock Vina or similar) and ABFE estimates? Given that Boltz-2 is cited as “near-FEP accuracy” in benchmarks, please provide specifics on this for the chemical space relevant to your study.
>
> **A1**: Given that Boltz-2 is trained on publicly available experimental binding measurement datasets like BindingDB, we observe that the correlation between Boltz-2 and experimental measurements is not very useful. However, what we do believe to be useful is the correlation between Boltz-2 and ABFE scores, which we do report in section 4.1. We find much more value in the direct comparison of Boltz-2 and ABFE that includes out-of-distribution ligands that Boltz-2 wouldn’t have seen during training, rather than the biased comparison of Boltz-2 and experimental affinities from BindingDB that were almost certainly within the Boltz-2 training set. We affirm the practical implications of our workflow with the strong correlation between Boltz-2 and ABFE, as it directly implies that Boltz-2 is "near-FEP accuracy" for our desired targets, and is thus likely to translate relatively well to experimental affinities, even in out-of-distribution scenarios (by virtue of its correlation to the established accuracy of free energy methods).
>
> >**Q2**: What are the practical implications of your workflow for medicinal-chemistry decision-making (hit-to-lead, lead optimisation)?
>
> **A2**: We consider the main practical advantages of our method to be for lead optimization for targets with some known binders / structures. While the original MOLLEO algorithm might have demonstrated strong AutoDock results for ZINC 250K structures, the primary practical advantages of our method come from fully exploiting known structures and the higher correlation Boltz-2 oracle to generate molecules that might actually have realistic chances of being strong binders in an experimental setting. The key advantage in this work comes from the focus on real-world experimental activity and ABFE validation, which makes MOLLEO+ a much more practical method than MOLLEO within a real-world drug discovery funnel.
>
> >**Q3**: On ABFE protocol. Could you provide full details on how many independent replicas per ligand were run, the resulting uncertainties, and convergence diagnostics?
>
> **A3**: We provide information about our ABFE setup in Appendix A. We performed 1 independent replica per ligand, with an observed average standard deviation in the predicted binding affinity of 1.35 kcal/mol, as determined by the ABFE software after a single run. For ligands yielded from Boltz-2 vs AutoDock oracles, we used the predicted pose from either oracle as the starting pose for each respective ABFE calculation. We believe that this is the most unbiased way to perform the ABFE calculation, since in order to give each ligand generated from each oracle the most favorable and fair setup in ABFE, we must use the corresponding pose from the oracle that determined the ligand to be strong. Otherwise, using the same oracle-predicted pose (e.g. Boltz-2) for all generated ligands would introduce bias toward only the ligands that were generated through the MOLLEO run using that oracle. We do not observe any steric clashes or significant conflicts in any ABFE simulations, suggesting that the poses are all initialized reasonably and that there is no particular bias toward any setup.
>
> >**Q4**: On SFT dataset quality. How did you verify chemical validity and label fidelity of the semi-synthetic dataset?...Please provide ablation results isolating the effect of SFT only
>
> **A4**: Since our SFT dataset is constructed from chains of BindingDB ligands that we know to be experimentally synthesized and tested, the dataset is guaranteed to have full chemical validity in its input and output molecules. It is not purely synthetic, and thus we do not discard any edits because we are certain that the "output molecule" (which is always just another ligand from BindingDB) is valid and plausible. Furthermore, our results in Table 3 on the effects of SFT are entirely isolated; the starting pool and oracle are fixed between the no-SFT and SFT runs, both using Boltz-2 and BindingDB. We don't believe a comparison on the original MOLLEO setup (docking and ZINC 250K) to be useful in this case, as we are primarily interested in demonstrating the relative gain in affinity from the SFT process, which is clear in this environment where all other factors are held constant.

---

> ### Author Response · Authors · 2025-12-03
> **Rebuttal by Authors (Part 2)**
>
> >**Q5**: On novelty and chemical novelty. Could you expand novelty analysis to scaffold-level novelty, and maximum similarity to any known binder across BindingDB or ChEMBL?
>
> **A5**: We have updated section 4.3 in the manuscript to include further analysis of novelty, including two different similarity thresholds and a filter employing Murcko scaffold-level similarity. As stated within the paper analysis, we acknowledge that MOLLEO+ does suffer from decreased novelty overall. However, we do not believe this necessarily hinders the practical utility of the method, and argue that for our intended points of optimization (translation to real-world practicality), the novelty results we have shown with several filters are sufficient to justify its usefulness in scenarios such as lead optimization and beyond. Ultimately, we are looking for molecules that are very strong binders that are perhaps dissimilar enough to be considered novel; we don't believe that a hard threshold much tighter than Tanimoto < 0.4 is justified or necessary for many practical applications. Thus we are comfortable in claiming that MOLLEO+ generations are sufficiently novel enough for the primary purpose that we are trying to achieve (lead optimization).
>
> >**Q6**: On baselines and fairness. Please include control runs that isolate each change independently: (a) original LLM + docking oracle + ZINC start, (b) original LLM + Boltz-2 oracle + ZINC start, (c) original LLM + docking oracle + BindingDB start, (d) fine-tuned LLM + docking oracle + BindingDB start
>
> **A6**: Table 3 already includes control runs (a) and (b) mentioned, under "MOLLEO" and "MOLLEO (Boltz-2)" respectively. We have also updated the manuscript so that Table 3 includes control run (c), under "MOLLEO (BindingDB)". As for the fine-tuned results, please refer to **A4** about why we don't believe the same controls to be necessary for the SFT comparisons. All methods are carried out under equal compute and each include 3 runs with random seeds.
>
> >**Q7**: the claim of GPT-4.1-mini being stronger than GPT-4 is not substantiated
>
> **A7**: We agree with the reviewer that this claim is not fully substantiated, and we have removed this statement in the revised manuscript. The original claim was motivated mainly by sources from OpenAI [1] but we acknowledge that this is not generalizable to our problem statement and thus is not a concrete statement. We don't believe this changes anything in our results, as everything is executed with the same GPT 4.1 mini model and thus all relative comparisons are still valid.
>
> [1] Introducing GPT-4.1 in the API | openai. OpenAI. (n.d.). https://openai.com/index/gpt-4-1/
>
> >**Q8**: Statistical reporting is incomplete: confidence intervals and error bars are missing
>
> **A8**: We provide confidence intervals in the form of standard deviation for the most crucial metrics (binding affinity) in all tables. In the case of Table 1 when there is varying sample sizes, we also provide a p-value (p = 0.007) calculated through the independent t-test, which inherently has varying sample sizes built into its calculation, meaning that the statistically significant p-value should be interpreted as such regardless of varying sizes between the independent samples.
>
> We thank the reviewer for their time and consideration in this process.

---

### Author Response · Authors · 2025-12-03
**Author's Message for Area Chair**

We thank the area chair for their time and consideration in evaluating our work. We would like to leave a comment summarizing our conversation with the reviewers and addressing a few fundamental misunderstandings.

Several reviewers had concerns about the structural novelty of molecules generated by MOLLEO+. While we have expanded section 4.3 to help address these concerns, we believe that, at a fundamental level, reviewers have misinterpreted the primary use-case of our framework. The purpose of MOLLEO+ is to exploit known structures in an effective way in order to optimize highly accurate ABFE affinity scores that are often neglected in work within this area; in other words, our primary focus is lead optimization and adjacent drug discovery funnels. We believe that we sufficiently demonstrate proof of novelty such that this focus on lead optimization is not at all compromised by the small decrease in structural uniqueness from the original MOLLEO. Please see **A2** and **A5** under rebuttals for Reviewer UiFU, **A1** under rebuttals for Reviewer MezY, and **A4** under rebuttals for Reviewer 1zdh for further detail.

Reviewer gqGE raised several concerns about our methods, stating many of them to be trivial. While we respect this feedback, we strongly disagree with these statements. The reviewer states that it's not surprising that using Boltz-2 instead of AutoDock results in higher ABFE scores. However, we believe that this result is not at all obvious; while docking may have been designed for large-scale screening, it reports a binding affinity prediction in just the same way that Boltz-2 does. Boltz-2 was not necessarily trained with FEP correlation in mind, so there is no reason to believe that ABFE-Boltz should be a correlated relationship, whereas ABFE-AutoDock is an uncorrelated one (like the reviewer claims). We believe that the high correlation we observe between Boltz-2 and ABFE is not an obvious result, and reveals the effectiveness of Boltz-2 in discovering high quality ligands that has not been well explored in previous work. In particular, we believe that our insights in section 4.1 regarding the increase in ABFE from using Boltz-2 over AutoDock are especially valuable and crucially demonstrate a comparison that has not been explored by other generative frameworks. This may constitute a strong argument for changing the field-standard affinity predictor (docking) that much current work still takes to be an accurate measure of generative success.

Reviewer gqGE further argues that utilizing BindingDB in the MOLLEO+ is not interesting, and is aligned with how the original MOLLEO was meant to be used. We respectfully yet firmly disagree with this perspective. Our fundamental goal for the MOLLEO+ framework is to optimize the original algorithm toward a process that may actually be useful within a real-world drug discovery funnel. The original algorithm selected strong binders from ZINC 250K through AutoDock prediction, which inherently introduces a level of artificial quality into the starting population. In MOLLEO+, sampling from a real protein-ligand database through evaluation of real-world experimental affinities constitutes a complete shift in the end goal for the optimization process. By aligning our algorithm with experimentally-verified binders, we shift the algorithm toward an actually realistic lead-optimization pipeline, something that was completely not explored and not intended within the original MOLLEO framework. We believe that this shift from artificial predictions to verified experimental affinities is non-trivial and a much more novel contribution than the reviewer believes it is. Please see **A5** and **A6** under rebuttals for Reviewer gqGE for further detail on both previous points.

We would like to acknowledge small changes to our manuscript, particularly the values within several tables in section 4. This was a result of added more seeded runs for robustness, in response to concerns/questions by reviewer gqPE. We have also further added several new sections and results to address various concerns by reviewers, such as concerns about ligand chains raised by reviewer gqPE (Appendix C.1) or concerns about control runs raised by reviewer UiFu (Table 3).

Finally, we would like to raise concerns about the quality of review from Reviewer 1zdh. While we greatly appreciate their insights and comments, the depth of the points listed in their review raise concerns about the detail they took in evaluating our work. We generally believe that these observations resulted from a very surface-level evaluation of our work; in particular, the strengths section is almost copy-pasted from our Introduction bullet points, and all concerns raised in the weaknesses section are ones we have already directly discussed and addressed in our Results, Discussion, and Limitations sections (as well as the Appendix). We believe this surface-level evaluation may have resulted in an inaccurate final score.

---

### Meta-Review · Area_Chair_m3S9 · 2026-01-06

**Summary:**

The paper proposes MOLLEO+, improving the MOLLEO framework by: (1) replacing AutoDock with Boltz-2 as oracle, (2) using BindingDB-derived starting populations, and (3) fine-tuning LLMs on semi-synthetic datasets. Results on c-MET and BRD4 show improved binding affinity predictions.

**Reviewer Concerns:**

Addressed: Statistical reporting (3 seeds per experiment), ablation controls (Table 3), expanded novelty analysis with multiple similarity thresholds (Table 4), ligand chain visualization (Appendix C.1).

Outstanding: Limited novelty (contributions viewed as engineering improvements), narrow evaluation (2 targets, ABFE only for c-MET).

**Reviewer Scores:**

Score would have been maintained to be negative in overall due to the outstanding concerns. Overall, I feel like reviewers are lukewarm due to lack of technical novelty combined accompanied with marginal significance in the empirical results.

---

### Decision · Program_Chairs · 2026-01-26

Reject